# Residue Folding Degree—Relationship to Secondary Structure Categories and Use as Collective Variable

**DOI:** 10.3390/ijms222313042

**Published:** 2021-12-02

**Authors:** Vladimir Sladek, Ryuhei Harada, Yasuteru Shigeta

**Affiliations:** 1Institute of Chemistry, Slovak Academy of Sciences, 845 38 Bratislava, Slovakia; 2Center for Computational Sciences, University of Tsukuba, Tsukuba 305-8577, Ibaraki, Japan; ryuhei@ccs.tsukuba.ac.jp (R.H.); shigeta@ccs.tsukuba.ac.jp (Y.S.)

**Keywords:** protein folding, secondary structure, networks, collective variables, molecular dynamics

## Abstract

Recently, we have shown that the residue folding degree, a network-based measure of folded content in proteins, is able to capture backbone conformational transitions related to the formation of secondary structures in molecular dynamics (MD) simulations. In this work, we focus primarily on developing a collective variable (CV) for MD based on this residue-bound parameter to be able to trace the evolution of secondary structure in segments of the protein. We show that this CV can do just that and that the related energy profiles (potentials of mean force, PMF) and transition barriers are comparable to those found by others for particular events in the folding process of the model mini protein Trp-cage. Hence, we conclude that the relative segment folding degree (the newly proposed CV) is a computationally viable option to gain insight into the formation of secondary structures in protein dynamics. We also show that this CV can be directly used as a measure of the amount of α-helical content in a selected segment.

## 1. Introduction

The prediction and analysis of folding pathways of proteins with atomistic models and molecular dynamics (MD) is presently, despite recent progress in AI aided predictions [1], perhaps still the best approach to the folding problem [2,3,4]; in particular, the identification of the folding pathway and local minima corresponding to partially folded (transition) states. Since the conformational space accessed during the MD simulations is typically multidimensional, the use of specialised coordinates—collective variables (CV)—is indispensable to make the analysis more tractable to humans. CVs are often used in various MD protocols to either extend the sampled part of the conformational space [5] and/or to select rare events and use these to speed up the simulation [6]. Traditionally, root-mean-square displacement (RMSD), the radius of gyration, and/or some eigenvectors from principal component analysis (PCA) are often used to capture the complex conformational changes and reduce them to a one or few parameter CVs. These, however, are incapable of providing qualitative and, in particular, quantitative insight into the formation of secondary structure (SS). Some CVs are designed to detect and quantify the amount of specific SS, for example, the (alpha) helical content CV as defined in NAMD [7] and beta strand content [8].

Recently, we have presented some findings about the relationship of the (residue) folding degree and protein dynamics [9]. In particular, we focused on the perspectives of the use of the folding degree to qualitatively and quantitatively measure the amount of folded content in proteins in molecular dynamics (MD) simulations of protein folding.

First, let us recall the origins of the folding degree. Introduced by Estrada [10,11], the folding degree was conceptualised as a one-parameter descriptor characterizing the compactness of a molecular structure. It was later re-defined as a metric relatable to the amount of folded structure in proteins [12,13]. The idea is quite elegant and relies on the fact that the backbone dihedral angles ψ, ω and φ, are confined to certain intervals in folded native structures. For rigorous definition, see the original works [10,11,12,13] or our paper [9]. For the sake of brevity we reproduce only the basic facts and definitions; however, a very recent review paper by Estrada uncovers the many facets of the use of this, and related, metrics [14].

The (global/average) folding degree is a parameter characterising the whole protein. It is defined via the spectrum of the adjacency matrix A, representing the third line graph of the (path) graph in which nodes correspond to the sequence of backbone atoms N, C${}_{\alpha }$, and C. The diagonal elements of the adjacency matrix, which are normally zeros, are replaced by the cosine of the respective dihedral angle ψ, ω or φ. This is because the vertices in this third line graph correspond to these dihedral angles. The adjacency matrix is a square matrix of size 3(N−1) for a *N* residue protein. The average folding degree $\langle C_{S}\rangle$ is calculated like the subgraph centrality [15], albeit for the weighting of A.
(1)$$\langle C_{S}\rangle =\frac{1}{3(N-1)}\sum_{i=1}^{3(N-1)}C_{S}^{i}\;,\;\mathrm{where}\;C_{S}^{i}=\sum_{j=1}^{3(N-1)}{\left(v_{j}^{i}\right)}^{2}e^{\lambda_{j}}.$$

The quantities $C_{S}^{i}$ are contributions of the *i*th vertex defined via the *i*th component of the 3(N−1) eigenvalues and eigenvectors of A; $\lambda_{j}$ and $v_{j}$, respectively. Since typically both ψ and φ can be defined for non-terminal amino acids, the residue folding degree [13] $RC_{S}$ of these amino acids can be defined as the sum of the corresponding $C_{S}^{i}$ values. If k=1,…,N is labelling the residues, and i=1,...,3(N−1) is labelling the vertices (backbone dihedrals), as in Equation (Equation 1), then $RC_{S}^{k}=C_{S}^{3(k-1)}+C_{S}^{3(k-1)+1}$, see also Appendix A.

This work can be considered a follow up on our first paper concerned with this topic [9]. In particular, we shall briefly revisit the scaling issue of calculating the residue folding degree from the whole matrix A or its sub blocks. Next, we try to establish a connection between $RC_{S}^{k}$ values and categories of protein secondary structure such as defined by the DSSP and STRIDE algorithms. Finally, we define the segment folding degree $RC_{S}^{\kappa }$ values for protein segments κ (parts of the sequence) and explore its use as collective variables to study potential energy surfaces from molecular dynamics simulations.

## 2. Results and Discussions

### 2.1. Calculation of the Residue Folding Degree

We have shown in our previous paper on this topic [9] that the residue folding degree $RC_{S}^{k}$ is well defined by the backbone dihedral angles φ and ψ at residue *k*. Hence, we have shown that it is not necessary to calculate the eigenvalues and eigenvectors of the full adjacency matrix A, rather the calculation from its 4×4 sub blocks is sufficient and introduces an error <1% compared to the calculation from the large matrix A. Equation (Equation 2) demonstrates this idea. The 4×4 sub blocks are in the larger blue rectangles and the corresponding φ, ψ angles in the smaller red ones.

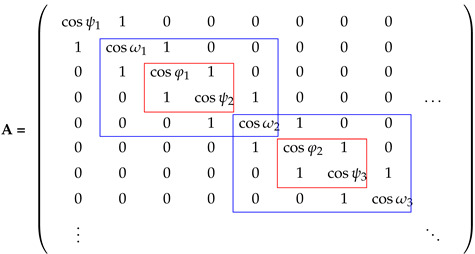
(2)

What we did not discuss in our previous paper is the computational time savings related to this block-wise calculation of the residue folding degree. To do so, we performed a series of calculations with varying size *n* of a fictional protein with up to 10${}^{4}$ residues; (3(n−1) is the size of the square matrix A). The results are displayed in Figure 1.

Evidently, the block-wise calculation is significantly more time saving. The absolutely dominant factor in this calculation is the evaluation of the spectrum of the adjacency matrix. We used the implementation of the python/numpy function like this: e_vals, e_vecs = np.linalg.eigh(A). As expected, the time costs rise linearly for the block-wise calculation, as it is essentially repeating *n* times the calculation of the spectrum of a 4×4 matrix. On the other hand, the calculation on the large matrix scales much faster than linearly. Some power law scaling can be anticipated based on the figure. Therefore, we recommend the calculation via the sub block method and do so for the remainder of this work.

Since the $RC_{S}^{k}$ value for the *k*th residue is so well defined by the φ, ψ at this residue, it makes sense to draw a Ramachandran-like plot of the residue folding degree, see Figure 2.

### 2.2. $RC_{S}$ and SS Categories

As we could see in Figure 2, the $RC_{S}$ values form characteristic patterns in a Ramachandran-like plot. It is well known that different SS categories also tend to have characteristic φ, ψ distributions in such plots. We have previously shown that, for these reasons, $RC_{S}$ can be used to differentiate between some of the SS categories; we used it to distinguish helical and β sheet/loop [9]. We did not however consider all the different SS categories and we did not use statistically significant samples to verify this. We attempt to do so now.

As mentioned in the Methods section, we used the database PolyprOnline with some >24,000 entries. We used the Bio.PDB [16] package to download and read them into our code. After each structure was successfully read we employed the DSSP, DSSP 4 and STRIDE algorithms to obtain the SS for residues in the structure. Additionally, we calculated the $RC_{S}^{k}$ values for the residues. The statistics of $RC_{S}^{k}$ corresponding to the SS categories were subsequently performed only if all the calculations finished without errors. The total number of such calculations/structures was 5392 with a total number of 1,299,653 residues of which 1,287,287 were standard, non-terminal amino acid residues. We considered this sample size sufficient for our purposes. A list of the PDB ID’s used in our analysis can be found at the end of the Appendix A.

First, as a side note, we consider the agreement of the SS categorisation between the three methods of choice. One can notice that the agreement between the “newer” DSSP 4 and the original DSSP is exceptionally high—besides the difference in the P category that is not defined in DSSP, only three of the remaining residues are classified differently. The agreement to STRIDE is also good, although there is a difference of some 4.3% in H, 4.6% in G, 96% in I, 2.8% in E, 0.6% in B, 79% in T, and 1.1% in the C category, when DSSP 4 is considered the reference. These findings, based on the last column in Table 1 and Appendix A, seem to be in agreement with other published results [17,18,19]. Apparently, STRIDE severely underestimates the π-helix conformation (I) compared to DSSP/DSSP 4 (visible also in Appendix A). In addition, the category T in STRIDE seems to comprise the T, S, C, and P categories of DSSP 4. A more detailed breakdown can be found in Equations (1) and (2) in the Appendix A. Therein, one can see which SS categories of DSSP and STRIDE correspond to DSSP 4 categories and *vice versa*. Finally, we show the statistics of amino acid distribution for each SS category. As expected, the category P, as defined by DSSP 4, is clearly dominated by proline, which can be viewed as both a test of the algorithms and our code correctness, see Appendix A.

The concluding message of this short analysis is that we (should) try to find characteristic $RC_{S}$ for both DSSP 4 and STRIDE categories, as they differ form each other.

Next, let us take a look at some typical $RC_{S}$ and φ, ψ values corresponding to the DSSP 4 categories compiled in Table 1. Similar data for all three algorithms are in Appendix A. Figure 3 and Appendix A show us that defining the secondary structure solely by $RC_{S}$ is rather impossible (at least to the categories commonly used by DSSP and/or STRIDE). We could relatively easily distinguish between helical structure (H, G, I) and beta structure (B, E for bridge and extended parts), as we did previously [9]. The hydrogen bonded turn (T) attains similar $RC_{S}$ values to the values for helical structures. This is however less clear/evident for the STRIDE category T, as it probably also contains part of the structures classified as S by DSSP. This can be seen from Figure 3 and Appendix A, where the distribution probability is (almost) unimodal for T defined by DSSP/DSSP 4, but is bimodal for STRIDE. For DSSP/DSSP 4, a part of the S (bend) structures attains $RC_{S}$ values very similar to the helical ones and T. Interestingly, the probability distributions (see violin plots) are close to centre-unimodal for H, G, I (and P). The probability distribution is not unimodal for the I category in STRIDE. Bear in mind that this category is the least populous of all. The probability kernel for E is also unimodal, but with the mode shifted to one side. The remaining ones, B, S, and C are multimodal. Neglecting the C category on the basis of not being a real category, rather a grouping of all those structures not fitting in the other categories, we can say that only the S category has three separate modes. Based on this, we can say that, if desired, the $RC_{S}$ could be used as an additional parameter to introduce a finer subdivision to the S category. One discriminating value for the residue folding degree could be close to the median value if two subcategories to S would be defined. More accurately, it should be at $RC_{S}=5$ if we look at the histogram in Appendix A. The values below the median can be further divided roughly at $Q_{1}$, or around $RC_{S}=2.6$. However, this second subdivision is less clear than the first one.

In conclusion, residue folding degree by itself is not suited to identifying classical SS categories. It can, however, be useful for discriminating between roughly what we could call bend and straight conformations. Into the bend we can more or less assign the categories H, G, I (helices), T (turns) and part of S (bend). The straight conformation would comprise E, B (β-sheet), P (PPII helix) and part of S. The majority of C would also fall here but, as mentioned, it is not a well defined category. The reason for only this rough classification of conformation via $RC_{S}$ is the same reason algorithms, such as DSSP and STRIDE, do not rely solely on the φ, ψ values in defining their respective categories. Several of the various SS categories share the φ, ψ space they occupy and therefore additional criteria had to be considered. To emphasize this point, we have drawn Ramachandran-like probability density histograms for each of the SS categories and relate them to typical $RC_{S}$ values, Appendix A. These demonstrate quite clearly that H, G, I, T and part of S conformations occupy the same or a very similar subspace to that of φ, ψ values. It also explains the bimodal nature of the distribution of $RC_{S}$ for the S category. Additionally, it visually supports the fact that category H has the narrowest $RC_{S}$ distribution of all the SS categories as it apparently occupies the narrowest interval of φ, ψ. Similarly, the P category has a narrow distribution of $RC_{S}$ as it occupies the most confined φ, ψ subspace per definition [20].

### 2.3. $RC_{S}$ and Helical Content

It is well known that protein structure has several levels, most notably the primary, secondary, tertiary and quaternary. The formation of all of these can be observed via molecular dynamics. One can use specialised collective variables such as the (alpha) helical content as defined in NAMD [7] and/or beta strand content [8] to measure the progress of secondary structure formation.

We used the α-helical content in the past [9] to characterise the folding process of deca-alanine (Ala${}_{10}$). Typically, these CVs can be applied to a smaller part/segment of the protein. Let us define the segment κ as a consecutive series of amino acids in the protein sequence. A motif shall be a segment with uniform SS category, that is, all residues in a motif fall under one SS category, but in a segment not necessarily. Motivated by the narrow distribution of $RC_{S}$ values associated with the H category, we attempt to define the relative (alpha) helical content of a protein segment by this quantity. Inspired by the NAMD formula we calculate the relative value for each residue k∈κ within the segment as:(3)$$RC_{S}^{k,rel}=\frac{1-{\left[\left(RC_{S}^{k}-RC_{S}^{ref}\right)/RC_{S}^{tol}\right]}^{m}}{1-{\left[\left(RC_{S}^{k}-RC_{S}^{ref}\right)/RC_{S}^{tol}\right]}^{n}},\;SC_{S}^{\kappa ,rel}=\frac{1}{\left|\kappa \right|}\sum_{k\in \kappa }RC_{S}^{k,rel}.$$

We use the exponents m=2, n=4, as these are also default in the NAMD equation to calculate alpha helical content, although one can use other values as well. The reference value is $RC_{S}^{ref}=7.273$ and the tolerance is $RC_{S}^{tol}=0.421$, which is the average residue folding degree value and its’ standard deviation for the H category as per the DSSP 4 rules, see also Table 1. The relative helical content in the segment κ is then calculated as the mean of the relative residue folding degree values $RC_{S}^{k,rel}$. We can denote this quantity as $SC_{S}^{\kappa ,rel}$.

To test our proposed measure of (alpha) helical content, we calculate these quantities for the trajectory of our MD simulation of folding for Ala${}_{10}$ published in [9]. Figure 4 depicts the comparison of the helical content calculated via Equation (Equation 3) and the NAMD equations. We can see that the general agreement is quite good for the snapshots and both values follow the same/a very similar pattern. The helicality content based on the $RC_{S}^{\kappa ,rel}$ seems to (almost systematically) attain somewhat lower values compared to the ones based on the NAMD formula. To confirm this we present the correlation of the two values in Figure 4b. Clearly, the correlation is satisfactory (R>0.9) and the systematic shift seems to be just under 0.1 (a=−0.0958).

Hence, we can conclude that the helical content calculated via Equation (Equation 3) can be used as a substitute for the NAMD collective variable.

One possible critique of this approach could be the fact that the residue folding degree interval, see Figure 3a, for α-helix (H) overlaps with those for 3–10 helix (G), partially π-helix (I), turn (T) and a part of bend (S). Hence, it can be argued that the identification of the H category may not be reliable/exclusive enough as it may include any of the mentioned SS categories. While strictly speaking, this argument is valid, it is not unique to the helicality CV based on the residue folding degree. The NAMD CV is defined by the angle of three consecutive $C_{\alpha }$ atoms ($C_{\alpha }^{k},C_{\alpha }^{k+1},C_{\alpha }^{k+2}$) and the distance between $O^{k},N^{k+4}$. One possible counterargument may be that sensible selection of the segments may avoid such a problem. Firstly, the selected segment should probably not be too short, as it is known that α-helix is characterised by the backbone interactions of residues k+4,k via the H-bond between NH and CO groups. The 3–10 helix is characterised by such k+3,k interactions and the π-helix by k+5,k H-bonds. Hence, to expect and quantify helical content in segment shorter than say 4 or 5 residues is atypical and the segment will more likely consist of turns an/or bends. To support this assumption, we present a statistical breakdown of motif lengths found in the studied reference set of PDB structures. It can be seen in Appendix A in the form of histograms. The most frequently occurring length of H motifs is 4 residue long with 10 being the second most frequent. On the other hand, turns (T) and bends (part of S) are considerably shorter with maxima at 2 and 1 residue length, respectively. This confirms our assumption. The G motif seems to be mostly of length 3 residues. Therefore, for longer segments it should be far more likely to quantify one of the helical SS categories H or I. The I motifs tend to be most frequently of length 5 and 6 residues. Let us look at whether the relative segment folding degree as defined by Equation (Equation 3) can differentiate between H and I. The mean value of the residue folding degree for I, $RC_{S}=6.516$, is outside the boundary of the interval defined as $RC_{S}=7.273\pm 0.421$ which is used as reference in Equation (Equation 3). Ergo, a segment of say length 5 or 6 residues that would be an I motif would not attain $SC_{S}^{\kappa ,rel}$ values close to 1, however a H motif would. In fact, the relative residue folding degree for a residue conforming to the "ideal" I motif (i.e., its’ $RC_{S}=6.516$) is only about $RC_{S}^{k,rel}=0.24$. Of course, for a pure I motif of any length this would also be its $SC_{S}^{\kappa ,rel}$ value. Table 2 contains such theoretical values for all pure motifs as classified by DSSP 4 SS categories. Therefore, the concluding remark of this discussion is that the helicality content defined via the relative segment folding degree is a viable alternative to the similar NAMD collective variable.

### 2.4. $SC_{S}^{\kappa ,rel}$ as a General Collective Variable

We have shown so far that we can quantify the helical content in a segment by $SC_{S}^{\kappa ,rel}$. We used the mean residue folding degree of the secondary structure category H (as by DSSP 4) as $RC_{S}^{ref}$ and the associated standard deviation as $RC_{S}^{tol}$ in Equation (Equation 3) for that purpose. We shall examine the possibility of whether the concept of relative segment folding degree $SC_{S}^{\kappa ,rel}$ can be generalised with the use of other (arbitrary) values for $RC_{S}^{ref}$ and $RC_{S}^{tol}$ in the following section. We will use the MD folding trajectory of the tryptophan cage (Trp-cage) as our model. The segment definition, as used in our work, and their correspondence to motifs in one (the first) of the experimental conformations of the structure with PDB ID: 1l2y [21] are depicted in Figure 5. Overlooking the fact that the motifs do not agree perfectly across the different SS categorisation algorithms, we see that neither the segments do not coincide with any of the motif ideally. This is done intentionally and is to no detriment to our purpose, as the presented structure is only one of the published snapshots.

The idea to generalise the use of $SC_{S}^{\kappa ,rel}$ is rather straightforward. First we should acknowledge that we use this quantity as a similarity measure between a given structure and a reference structure (as was the SS cat. H for helicality). We still continue to use Equation (Equation 3) to calculate the similarity. However, here we can also use different reference structures via different $RC_{S}^{ref}$ and $RC_{S}^{tol}$ values. The first step is to select a reference structure and a residue sequence(s) within the structure as our segment(s), see Figure 5. Next we must select the reference values. Several options come to mind. One of the simplest is to calculate the $RC_{S}^{k}$ for the residues in the segment and take their mean as the reference value within the given segment. The results for this kind of analysis are depicted in Figure 6. To calculate the $RC_{S}$ values you can use, for example, our program [22] pyProGA (https://gitlab.com/Vlado_S/pyproga, accessed on 24 July 2021). Since we take the difference of $RC_{S}^{ref}$ and $RC_{S}^{k}$ for each residue *k* in the segment (see Equation (Equation 3)), this approach should work best if the residues in the segments have similar $RC_{S}^{k}$ values to begin with. This is to assure that $SC_{S}^{\kappa ,rel}\to 1$ when the conformation of the segment is similar to the conformation of the reference structure. In other words, $RC_{S}^{k}-RC_{S}^{ref}\to 0$ for each residue in the segment. For a better picture, how $RC_{S}^{k,rel}$ depends on $RC_{S}^{k}$ see Appendix A. This figure also shows that, as expected, the larger $RC_{S}^{tol}$ values lead to a less sharp peaks. For the helicality content, we used the $RC_{S}^{tol}=0.421$, which is the standard deviation of the mean $RC_{S}$ value associated with the H category as by DSSP 4. There is no reason to use this value for structures with arbitrary reference $RC_{S}^{ref}$. Hence, we explored the effect of setting $RC_{S}^{tol}=0.5$ and 1.0 on the energetic surfaces representing folding processes in the Trp-cage mini protein. Figure 6 shows that the choice of $RC_{S}^{tol}=1$ leads to clearer PMF plots with better separation of local minima and the folded/unfolded structures. Appendix A yields to similar conclusions for the remaining CVs. We will come back to how to read these plots.

There are alternative ways to define the reference values $RC_{S}^{ref}$. Previously, we used one value for each residuum within the segment (the average $RC_{S}$ within the given segment). We can use also omit the average and use the $RC_{S}$ for each residuum as in the reference structure. PMF plots for this case are shown in Appendix A. Lastly, in a similar fashion, we can use the characteristic $RC_{S}$ value (see Table 1) associated with the SS category of the residuum in the reference structure. Such PMF plots are shown in Appendix A. In both cases, we can achieve better resolution of the states when looser $RC_{S}^{tol}$ values are applied. We conclude this on the basis of Appendix A, where we compare the PMFs with $RC_{S}^{tol}$ being either the standard deviation of the characteristic $RC_{S}$ value (see Table 1) associated with the SS category of the residuum in the reference structure or a fixed value of $RC_{S}^{tol}=1$. Appendix A compares the individual $SC_{S}^{\kappa ,rel}$ values for three different reference $RC_{S}^{ref}$ values. This is done mainly to check whether the numeric value of the CV depends on the particular choice of the reference for each snapshot. We see that they do largely follow the same major trends, albeit with some differences. In our particular application, the $SC_{S}^{(2-9),rel}$ seems to be a useful CV with consistent values regardless of the choice of the reference values. On the other hand, $SC_{S}^{(10-11),rel}$ seems to work when we use the average segment $RC_{S}$ and/or the $RC_{S}$ of each residuum, but is less useful if one chooses the characteristic $RC_{S}$ for the SS category of the residue in the reference frame. All in all, our data suggest that with the variety of reference choice one can probably select such CV, which can competently describe the evolution of secondary structure in the MD simulation. The choice of $RC_{S}^{tol}$ remains somewhat arbitrary as we do not see any rigorous way how to find some “optimal” value. Nonetheless, $RC_{S}^{tol}=1$ seems to yield PMFs with energy barriers comparable to barriers of partial folding processes found by other researchers as we show in the next part.

This brings us to the last point—the interpretation of these plots along with some concluding remarks. To do this, it is advantageous to pinpoint what the residue and segment folding degrees actually tell us. Firstly, it is a “data reduction” technique in the sense that, for each residuum, one can reduce the information about the backbone conformation from two parameters (φ, ψ) to one parameter with certain degree of accuracy. Secondly, the segment folding degree can be viewed in two ways, albeit, in both ways it acts as further data reduction in so far as individual residue folding degrees (multiple parameters) are replaced with the segment folding degree—a single parameter. On one hand, it can be used to asses/categorise the secondary structure. We have shown that it cannot (and is not intended to) replace the established SS categorisation algorithms such as DSSP and/or STRIDE. It can, however, be used for a basic differentiation between helical/turn and extended/β-strand-like conformations. On the other hand, the relative segment folding degree $SC_{S}^{\kappa ,rel}$ can be used effectively to quantify the amount of α-helical content in a segment. This is mainly due to the tight standard deviation of the characteristic residue folding degree corresponding to α-helices. In other words, $SC_{S}$ can measure the similarity of a given structure to α-helices rather accurately. By choosing another, arbitrary reference structure, one can quantitatively measure the similarity between the reference and other structures. Hence, in this sense $SC_{S}^{\kappa ,rel}$ can act very similarly to the root-mean-square displacement (RMSD) collective variable. Yet, $SC_{S}$ is more specific, as it compares the similarity in secondary structure. With this in mind, if we look at, for example, Figure 6b,c, we see that there are several local minima. One (C) with $SC_{S}^{(2-9),rel}\approx 0.9$ and $SC_{S}^{(12-15),rel}\approx 0.7$. This one corresponds to the folded structure with formed secondary structure (in the two respective segments). Of course the particular numeric values of $SC_{S}$ depends on the choice of the reference structure. We know that this minimum belongs to the folded structure because in our case we took the reference to be one of the NMR snapshots of the folded Trp-cage structure (model number 1 in the PDB ID 1l2y [21]). The minima at $SC_{S}^{(2-9),rel}\approx 0.3$ and $SC_{S}^{(12-15),rel}\approx 0.05$ (A) as well as at $SC_{S}^{(2-9),rel}\approx 0.3$ and $SC_{S}^{(12-15),rel}\approx 0.3$ (B) correspond to structures with secondary structure unformed or partially formed. This tells us also that the formation of the helix in the segment (2–9) is quite simple in so far as there are only two majorly populated states along the $SC_{S}^{(2-9),rel}$ axis (both minima A and B have approx. the same $SC_{S}^{(2-9),rel}$ values). Along the $SC_{S}^{(12-15),rel}$ axis we encounter three noteworthy minima, suggesting that the residues in the respective segment adopt multiple backbone conformations in the folding process. We can analyse the other plots in Appendix A in a similar fashion. For example, we see that the secondary structure in the segment (16–19) adopts the close-to reference structure rather early in the process and remains stable. In particular, Appendix A confirms this with $SC_{S}^{(16-19),rel}≳0.8$.

The concept of relative segment folding degree lends itself rather well to the creation of minimum energy paths (MEP), as can be seen in Figure 6 and Appendix A. We calculated the MEP with the MEPSA tool [23,24]. The energetic barriers separating the states (secondary structure unfolded and folded) are as much as 1.8, 1.7, and 1 kcal mol${}^{-1}$. Streicher and Makhatadze report experimental overall (un)folding free energy of Trp-cage at 25 °C of some ΔG=0.76±0.05 kcal mol${}^{-1}$ [25], stating that the mini protein is marginally stable at that temperature. Other experimental studies exist [26]. Computational assays on the Trp-cage folding report various free energy barriers for particular/partial processes/coordinates; for example, Zhou [27] finds a salt bridge stabilised intermediate state with a barrier ≈1 kcal mol${}^{-1}$. Juraszek and Bolhuis find transition barriers up to ≈3–4 kT, with kT corresponding to some 0.596 kcal mol${}^{-1}$ at 300 K, see also Figure 6b for comparison. Other researchers have found similar results [28,29,30]. In this respect, our trajectory and the associated PMFs seem sensible and the analysis of the process via the relative segment folding degree seems to bring additional insight into the stages of secondary structure formation. Recent findings have shown that (partial) formation of secondary structure in the early stages of protein folding may play important roles in the overall folding process [31]. We would like to emphasize that the residue/segment folding degrees are not meant to be a universal CV that can describe all structural transitions at liberty. Folding models, including the concept of the so-called molten globule state/phase [32], assume that the native secondary structure is partially formed in the earlier stages of folding. Subsequent processes involve the emergence and rearrangement of domains into the native tertiary structure. These processes often involve transitions over slow degrees of freedom with distinct structural intermediates. For a proper description of these, more involved MD simulation strategies are required (beyond one long continuous simulation) and the analysis of the trajectories via so-called Markov state models (MSM) [4,33] is a viable strategy. Specialised tools may be used for that purpose [34,35,36]. The work of Sidky et al. [28] presents such an analysis of the Trp-cage protein using state-free reversible VAMPnets (SRVs). These present an alternative to the time-lagged independent component analysis (TICA) commonly associated with MS models for protein folding [37,38]. This technique allows the identification of the slow degrees of motion that are subsequently used for clustering, and hence the identification of transition states. The authors state that [28]*“TICA has all but superseded structural clustering based on metrics such as minimum root-mean-square distance (RMSD) that tend to capture motions of high structural variance as opposed to the desired slowest motions”*. In this sense, we expect our CV to perform similarly to RMSD, as it is also based on the comparison of the secondary structure between a reference and the structures in the MD trajectory. That is, our CV is likely to capture motions with greater variance in the secondary structure. This may be considered a drawback if one wants to capture the slow transitions [39]. Such problems are often encountered in MD simulations of intrinsically disordered proteins (IDP) (in absence of their natural “partner” molecules) [39]. They can attain several “folded” conformations with similar (global) folding degrees, whereas the "true" folded state is based on crystal structures. Our proposed CV can differentiate the intermediates if they differ in secondary structure, that is, it would, as mentioned, identify structurally varied conformers, much like RMSD, although SS transitions will be more targeted. It should be noted that, in cases of IDPs, the selection of suitable CVs is problematic in general and one cannot (should not) rely on the simplest selection, for example, RMSD. Perhaps an advantage of our CV is that, unlike the RMSD, which has no upper bound, our CV spans the interval between zero and one. As our goal is to demonstrate a CV for the analysis of formation of SS, the analysis of slow modes is beyond our current scope. The usefulness of the relative segment folding degree as a coordinate to be used with TICA and/or SRV is a question open for further detailed analysis. In conclusion, we recommend the use of the $SC_{S}^{\kappa ,rel}$ for similar analyses, in the hope that they may prove useful for proteins other than these example cases as well.

## 3. Materials and Methods

### 3.1. Selection of the Protein Structure Database

In the fist part, we investigate the relationship between $RC_{S}^{k}$ values and secondary structure (SS) categories. Perhaps, the best known algorithms that define SS are the DSSP [40] and STRIDE [17] protocols, although others exist (e.g., SECSTR [41], DEFINE [42], P-CURVE [43], P-SEA [44], XTLSSTR [45], PSSC [18], VoTAP [46]). A good review on this subject is by Martin et al. [19]. We used both DSSP (from Ubuntu repo via sudo apt install dssp) and STRIDE (http://webclu.bio.wzw.tum.de/stride/, accessed on 24 July 2021). Typically, the categories in both DSSP and STRIDE are quite similar, although the agreement between the assignment to each category may differ [17,19]. The helical SS categories are H (α helix), G (3–10 helix), and I (π helix). The strands comprise the categories B (β sheet) [47,48] and E (extended). The remaining ones are T (turn), and C (coil). These categories are identical in both DSSP and STRIDE; however, in STRIDE, the category C is assigned to coils and “none of the remaining ones”, whereas DSSP assigns a blank to an undefined/irregular structure. Additionally, the DSSP protocol defines also the S (bend) category. In recent years, it became evident that neither DSSP nor STRIDE are able to distinguish the polyproline helix II (PPII) [49,50]. This was corrected in the latest iteration DSSP 4.0 protocol https://github.com/PDB-REDO/dssp (accessed on 24 July 2021), which defines the PPII helix as category P and it is a subset of the category C, meaning that PPII is assigned only to structures which in the older DSSP protocols would fall under C (blank) [20]. The algorithm was implemented into a code as part of the PDB-REDO project [51]. Since we also wanted to make use of this latest method, we chose the database compiled by the authors of the algorithm [20], which can be accessed from https://webs.iiitd.edu.in/raghava/ccpdb/collect.php (accessed on 24 July 2021) under the name PolyprOnline, or their web page https://www.dsimb.inserm.fr/dsimb_tools/polyproline/about.php (accessed on 24 July 2021). This database contains some 24,761 folded protein structures.

In this first part, where we focus on the relationship between $RC_{S}^{k}$ values and SS categories, we use only static protein structures from the PolyprOnline database. We would like to declare upfront that we do not intend/propose to use the residue folding degree as a substitute for SS defining algorithms. The dynamical aspects of stabilising effects of proteins SS are studied elsewhere [52,53,54].

### 3.2. Molecular Dynamics

In the later part, we investigate the use of the residue and segment folding degrees as collective variables to investigate molecular dynamics (MD) trajectories. We use two MD trajectories of the simulated protein folding of deca-alanine (Ala${}_{10}$, ten consecutively joined alanines) and the tryptophan cage (Trp-cage). Extended linear strands were taken as the starting structures. The peptide was solvated with TIP3P waters [55] in a rectangular box with an appropriate number of Cl${}^{-}$ ions added to neutralize the system. The chemical bonds were treated as rigid bodies with the LINCS algorithm [56]. Peptide termini were capped with ACE and NME groups in deca-alanine and with hydrogens in the Trp-cage. The water molecules were treated as rigid bodies with the SETTLE algorithm [57]. The temperature of the system was controlled with the modified Berendsen thermostat [58]. The pressure of the system was controlled with the Parrinello–Rahman method [58,59]. The electrostatic interactions were calculated with the particle mesh Ewald method using a real space cutoff of 10 Å [60]. The cutoff value for van der Waals interactions was set to 10 Å. To equilibrate the solvated system, a 100 ps MD simulation was initiated from the extended strand under NVT constraints (T=300 K) and 1 μs and 10 μs MD simulation was sequentially performed under NPT constraints (T=300 K and P=1 bar) for Ala${}_{10}$ and Trp-cage, respectively. Snapshots were recorded each 100 ps, so in total we analyzed $10^{4}$ snapshots for Ala${}_{10}$ and $10^{5}$ for Trp-cage. We observe the reversible folding and unfolding of the structures in this time frame. All the MD simulations were performed with the GPU version of the GROMACS 2018 package [61] using the AMBER ff14SB force field [62].

## Figures and Tables

**Figure 1 ijms-22-13042-f001:**
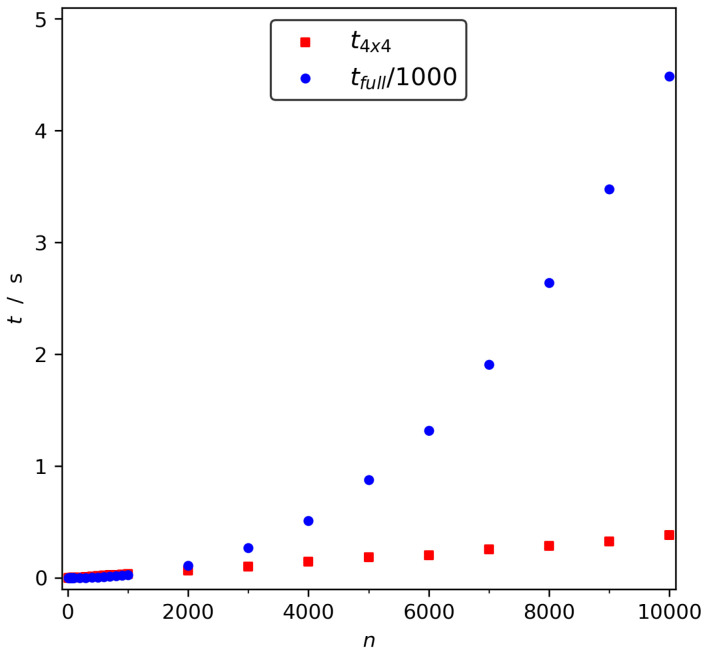
Time scaling for the calculation of $RC_{S}^{k=1...n}$ values either from the 3(n−1)×3(n−1) matrix A (blue dots) or from its’ 4×4 sub blocks (red squares). Notice that the plotted time for the former is divided by 1000 to fit the figure nicely.

**Figure 2 ijms-22-13042-f002:**
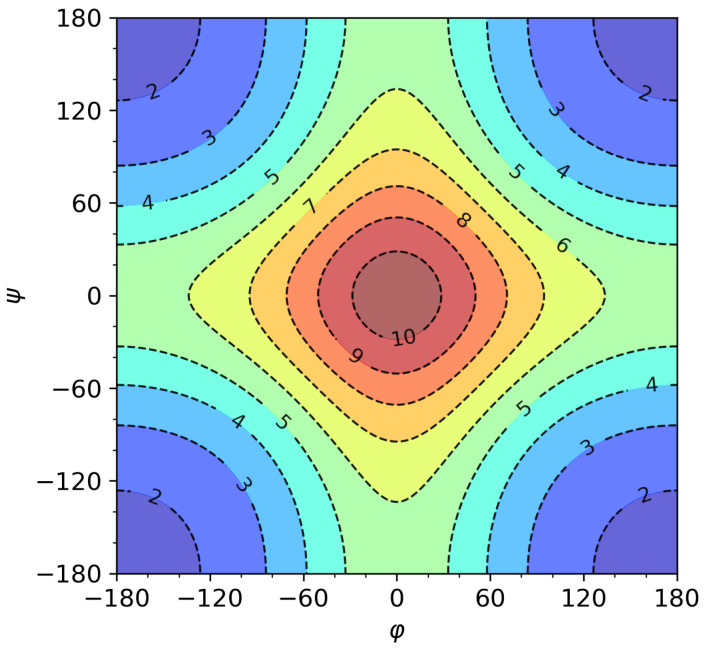
Theoretical $RC_{S}$ values as a function of φ, ψ calculated at a grid with step size of 0.75°. The contours show the numerical value.

**Figure 3 ijms-22-13042-f003:**
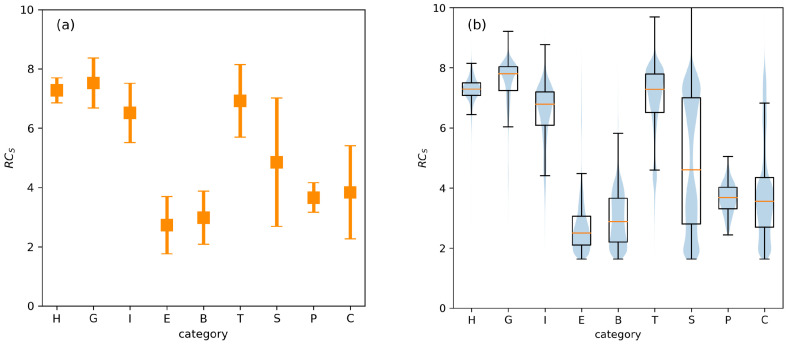
Typical residue folding degree values for SS categories from DSSP 4. The mean value and the standard deviations are shown in (**a**) and the densities and box plots in (**b**). The box plot also indicates the mean (orange line), the lower and upper quartiles $Q_{1}$, $Q_{3}$ and the whisker ends indicate statistical outliers at $Q_{1}-1.5\left(Q_{3}-Q_{1}\right)$ and $Q_{3}+1.5\left(Q_{3}-Q_{1}\right)$.

**Figure 4 ijms-22-13042-f004:**
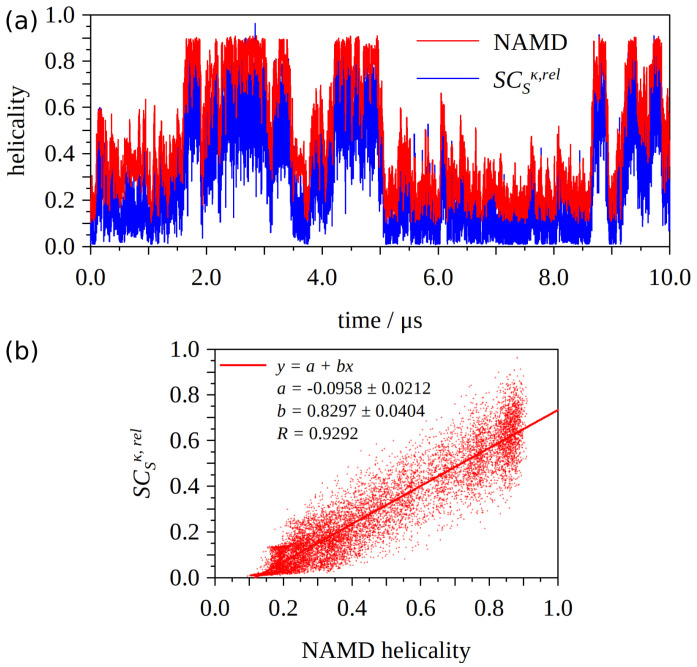
Helical content in (Ala${}_{10}$) calculated via Equation (Equation 3) and the NAMD equations (see the NAMD document COLLECTIVE VARIABLES MODULE Reference manual for NAMD, or [9]). Part (**a**) shows the values for recorded snapshots of the MD trajectory. Part (**b**) shows the correlation between the two quantities.

**Figure 5 ijms-22-13042-f005:**
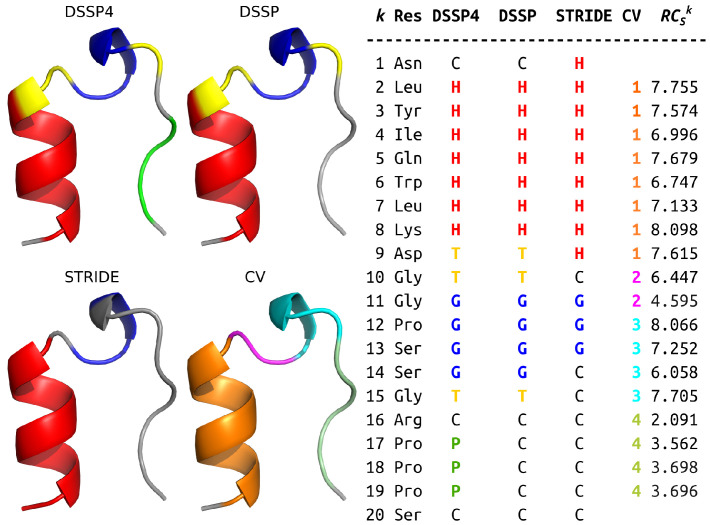
Secondary structure categories in Trp-cage (model 1 in PDB ID: 1l2y) as defined by various algorithms. The categories are designated a color and summarised in the table. The assignment of residues to segments, *ergo* to collective variables used in this work is also defined. The last column contains residue folding degree values for each residuum in the given structure. The averages of $RC_{S}$ for the segments are 7.450 for seg. 1 (res. 2–9), 5.521 for seg. 2 (res. 10–11), 7.270 for seg. 3 (res. 12–15), and 3.262 for seg. 4 (res. 16–19).

**Figure 6 ijms-22-13042-f006:**
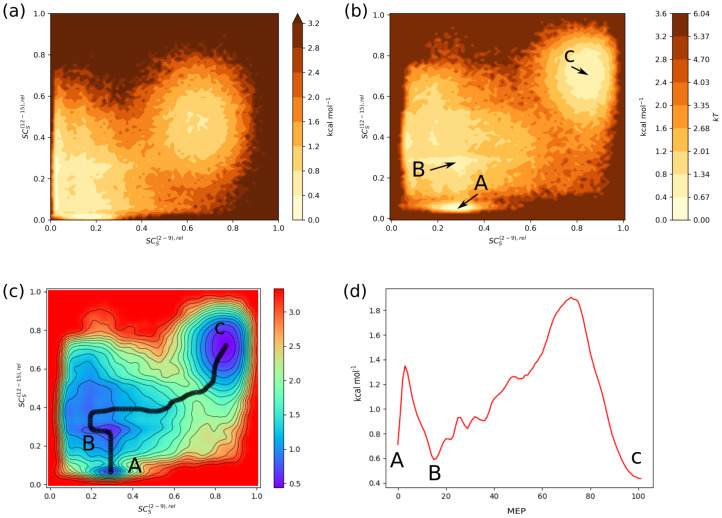
Potential of mean force (PMF) surfaces (a.k.a. free energy landscape, FEL) at 300 K for the folding process of the Trp-cage protein captured by two CVs; $SC_{S}^{(2-9),rel}$, and $SC_{S}^{(12-15),rel}$. See Figure 5 for CV and $RC_{S}^{ref}$ definition. In part (**a**) we used the tolerance $RC_{S}^{tol}=0.5$ and in part (**b**) $RC_{S}^{tol}=1$ was used. The resulting minimum energy path (MEP) connecting the unfolded (A), partially folded (B) and folded (C) states (**c**) relates to the latter surface. The energy profile along the MEP coordinate is shown in part (**d**). All energies in kcal mol${}^{-1}$.

**Table 1 ijms-22-13042-t001:** Typical $RC_{S}$ values for SS categories as defined by the DSSP 4 algorithm (i.e., incl. category P). The mean values with (sample) standard deviations in parentheses are given for $RC_{S}$ and the φ, ψ values. Count indicates how many residues these numbers were based on.

SS Cat.	φ	ψ	${\mathrm{RC}}_{S}$	Count
H	−64.70 (11.84)	−39.56 (11.39)	7.273 (0.421)	412,071
G	−66.05 (34.16)	−15.55 (29.33)	7.523 (0.840)	51,822
I	−79.15 (25.74)	−41.63 (20.41)	6.516 (0.999)	7086
E	−110.89 (42.62)	122.38 (58.10)	2.730 (0.966)	282,196
B	−96.98 (49.29)	122.90 (67.45)	2.976 (0.898)	15,416
T	−39.33 (70.25)	6.23 (51.49)	6.923 (1.223)	151,631
S	−69.26 (73.32)	44.16 (97.63)	4.850 (2.166)	113,536
P	−72.33 (13.05)	144.79 (13.85)	3.660 (0.498)	24,764
C	−82.97 (55.81)	97.00 (83.52)	3.835 (1.574)	218,948

**Table 2 ijms-22-13042-t002:** Typical $SC_{S}^{\kappa ,rel}$ values for ideal SS motifs as defined by the DSSP 4 algorithm, when the H category is the reference state. The “ideal” motif means that all residues in it attain mean $RC_{S}^{k}$ values for given SS category, see Table 1.

SS Cat.	H	G	I	E	B	T	S	P	C
$SC_{S}^{\kappa ,rel}$	1.00	0.74	0.24	0.01	0.01	0.59	0.03	0.01	0.01

## Data Availability

The data presented in this study are available in the Appendix A
*support_info.pdf* and our code(s) used for the analysis alongside the raw data are to be found here: https://gitlab.com/Vlado_S/folding-degree-cv-and-protein-secondary-structure, accessed on 14 October 2021.

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
