# Peer review of "Residue Folding Degree—Relationship to Secondary Structure Categories and Use as Collective Variable"

_ijms, 2021, doi:10.3390/ijms222313042_

Round 1

Reviewer 1 Report

The authors presented a manuscript focused on applying a network-based measure of folded content in proteins, the so-called "residue folding degree" to analyse the formation of secondary structure. The residue folding degree has been demonstrated to be useful in particular to measure the alpha helical content in a selected protein segment. Despite the manuscript may be of interest for a very specialized readership, it is well written and methodologically sound. However, I ask the authors to address the following points:

  • Since the beginning of the manuscript, the authors presented the residue folding degree as collective variable for MD simulations. I was really excited about this point, and then disappointed to see that they only use this parameter to analyze classical MD simulations. In order to demonstrate that a specific parameter is a good collective variable for a molecular event, the authors should demonstrate that this CV includes all the relevant slow degrees of freedom. Moreover, they should demonstrate that enhanced sampling based on this variable is able to correctly reproduce the free energy landscape of the protein. Instead, the authors have "only" demonstrated that the folding degree is a good parameters to analyse the dynamic of protein folding. I would deeply change the manuscript considering this point. Within this context, the title of the manuscript is also misleading.
  • Another concern is about the limitations of this parameter in correctly describing the folding dynamics. We can see only two examples (deca-alanine and the tryptophan cage) of the application of the folding degree, and it is difficult for the reader to understand the limits of the applied protocol for more general purposes.

Reviewer 2 Report

The paper ”Residue Folding Degree - Relationship to Secondary Structure Categories and use as Collective Variable”  from Sladek et al develops a collective variable for MD based on this residue-bound parameter to be able to trace the evolution of secondary  structure in segments of the protein. Futhermore the authors report that this CV can be directly used as a measure of the amount of α-helical content in a selected segment. 
The manuscript reports some interesting results and is well written. My suggestioni is that The manuscript is worthy of pubblication in the International Journal of Molecular Sciences in the present form.

Round 2

Reviewer 1 Report

The authors did not properly address my question about the limitations of this work, which is a fundamental aspect of any research study. The answer of the authors is that "using a larger / more complicated protein could potentially muddy the message". However, I did not explicitly ask any additional simulation, but I strongly ask the authors to make an effort in order a) to highlight the limits of their methodological approach, b) to contextualize these limitations in the scientific literature. Otherwise, the presented results are useless for the scientific community working in this field.

Round 3

Reviewer 1 Report

The authors addressed the Reviewer's comments